# MoB: Mixture of Block Transformer for Accelerating Video Generation with Dynamic Routing

## Abstract

Diffusion Transformers (DiTs) have demonstrated exceptional performance in high-fidelity image and video generation tasks. However, their iterative denoising process introduces substantial computational redundancy within Transformer modules, resulting in prohibitively high computational costs and slow inference speeds. Through comprehensive experimental analysis of existing DiTs, we reveal two key observations: (1) outputs of different Transformer blocks exhibit significant similarity during the denoising process, and (2) block-level redundancy varies dynamically across denoising timesteps. Based on these insights, we propose **Mixture of Blocks (MoB)**, the first framework to introduce block-level dynamic routing for DiT acceleration. The core innovation of MoB lies in a lightweight routing network that dynamically evaluates the importance of each Transformer block based on input prompts. At each denoising step, we propose the Ada-Top-$k$ mechanism which selects relevant blocks by using the k-th largest score as an adaptive threshold, avoiding the winner-take-all problem of traditional soft selection while eliminating 10-20% of redundant computations. To compensate for information loss from skipped blocks, we design a Block Cache mechanism that maintains generation quality by reusing intermediate feature differences from previous timesteps. Furthermore, MoB integrates adaptive timestep skipping and employs knowledge distillation to train the routing network, achieving enhanced inference efficiency and training stability. In addition, we evaluate its generalization ability on image generation tasks using Flux.1. Extensive experiments demonstrate that MoB achieves significant inference acceleration while preserving generation fidelity in both video and image generation tasks, consistently outperforming existing baseline methods in both efficiency and quality.

## 1 Introduction

Video generation models based on the Diffusion Transformer (DiT) architecture remain computationally expensive (He et al., 2024; Yuan et al., 2024a; Fei et al., 2025; Zhang et al., 2025). These models typically start from a random noise initialization and iteratively denoise over multiple steps. While effective in producing high-fidelity results, this multi-step sampling trajectory incurs substantial computational overhead and poses challenges for real-time or interactive applications. Consequently, accelerating inference in DiT-based models has become a critical research problem.

Early acceleration methods primarily targeted the diffusion sampling process itself, such as DDIM (Song et al., 2022) and DPM-Solver (Lu et al., 2022). Although these approaches reduce the number of sampling steps, they still require multiple iterations and cannot fundamentally eliminate structural redundancy. Distillation-based methods (Zhai et al., 2024) compress hundreds of sampling steps into only a few via a teacher–student framework, but the distillation process remains computationally demanding. Hardware-aware optimizations, such as FlashAttention (Dao et al., 2022), offer composable low-level speedups but provide limited cross-hardware portability and only constrained algorithmic improvements.

Beyond these, architecture-levelstrategies, such as pruning (Wu et al., 2024), low-rank decomposition (Hu et al., 2025), and dynamic routing (Sun et al., 2025; Shi et al., 2025; Xi et al., 2025)—di-

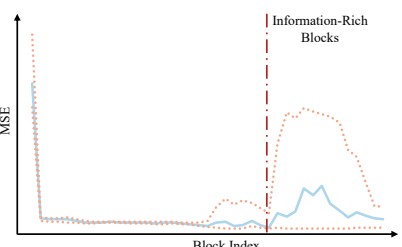 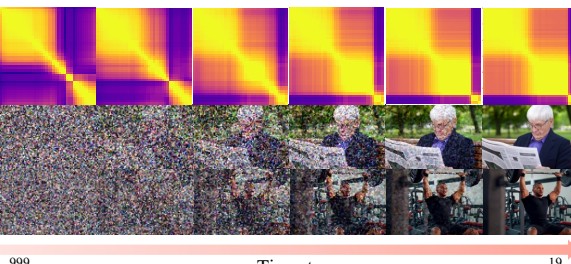

Figure 1: Block output analysis in CogVideoX-5B. **Left:** MSE between outputs obtained by skipping adjacent blocks. The blue solid line indicates the average MSE across all prompts, while the yellow dashed lines denote the maximum and minimum values. Blocks with higher MSE values are regarded as more informative, whereas those with lower MSE values can potentially be skipped without incurring significant performance degradation. **Right:** Cosine similarity between block outputs across timesteps. In the heatmap, both axes represent block indices, with brighter colors indicating higher similarity. The results highlight the presence of block-level redundancy and motivate the use of dynamic routing in MoB.

rectly optimize the generative model at the algorithmic level. Among structural optimization techniques, dynamic routing has typically been applied within Transformer blocks to specific components (e.g., attention (Jin et al., 2025a) in Figure 2(a) or MLP (Fei et al., 2024) in Figure 2(b)), motivated by its similarity to Mixture-of-Experts architectures. In contrast, block-level optimizations have largely relied on pruning or low-rank decomposition. However, these methods neglect the input sensitivity of entire Transformer blocks, thereby limiting the model's ability to adaptively control depth during inference.

Following the approach of Daniel Verdú (2024), which prunes blocks with small input–output mean squared error (MSE) values to reduce model size from 12B to 8B parameters, we conduct analogous experiments on CogVideoX-5B (Yang et al., 2025) using a randomly selected batch of prompts from the Lin et al. (2014). Specifically, $n$ prompts (e.g., 200) are input into CogVideoX-5B, and the generation process is manually terminated after specific Transformer blocks. As shown in Figure 1, for each case, we compute the MSE between the outputs of adjacent blocks. The averaged results, together with their upper and lower bounds across 200 measurements. To further assess the effect of timesteps on block outputs, we also record the outputs of all blocks at multiple timesteps and plot cosine similarity maps across blocks over different denoising stage.

Our findings reveal that, although the contribution of a given Transformer block varies across inputs, there consistently exist block pairs whose output differences remain minimal and, in some cases, negligible. Similar observations are reported in Daniel Verdú (2024), which shows that blocks with small input–output differences exert limited influence on the final generation quality. These results indicate that DiT-based video generation models exhibit substantial computational redundancy at the block level. Moreover, block similarity is relatively low in the early stages of denoising but becomes more pronounced in later stages. This suggests that block-level redundancy increases as denoising progresses, consistent with prior findings on block-level pruning (Zhao et al., 2025; Chen et al., 2024).

Motivated by these insights, we propose the **Mixture of Block Transformer (MoB)**, a DiT-based text-to-video generation framework that employs block-level dynamic routing. As illustrated in Figure 2(c), MoB explicitly selects and activates only the most relevant blocks, thereby reducing inference cost while preserving high-quality generation performance.

MoB introduces a dynamic routing network that computes block-wise relevance scores from input text features. These scores are processed through our **Ada-Top-$k$** mechanism, which uses the k-th largest score as an adaptive threshold rather than traditional competitive normalization. This rank-based selection strategy activates blocks whose scores exceed the threshold, effectively pruning redundant computations while avoiding the winner-take-all collapse inherent in softmax-based approaches. Concretely, MoB processes input text embeddings to generate relevance scores for each DiT block, then applies Ada-Top-$k$ to determine which blocks should be executed at each iteration of the denoising trajectory.

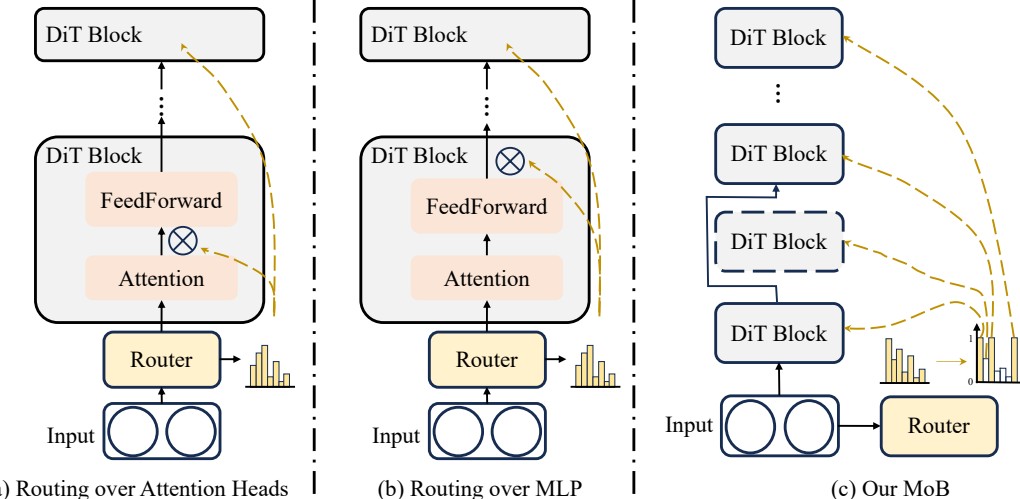

Figure 2: Utilization of three types of dynamic routing within the DiT framework.

To compensate for potential information loss from skipped blocks, we incorporate a **block cache module** that aggregates intermediate representations into a unified output. Furthermore, MoB adopts **knowledge distillation** and **load balancing** strategies, which improve training efficiency and enable effective fine-tuning on small-scale datasets, thereby enhancing the framework's practicality and deployability.

## 2 METHODOLOGY

We propose the **Mixture of Block Transformer (MoB)**, an optimization framework built upon the CogVideoX-5B architecture. MoB employs a routing network to dynamically select Transformer blocks, complemented by fine-grained design choices to address challenges in practical deployment.

Prior to the denoising process, the text embedding is passed through the routing network, where global pooling followed by a linear projection produces a vector of routing scores, each corresponding to a Transformer block. During inference, Ada-Top-$k$ strategy is proposed to select the indices of the $k$ most relevant blocks, which are then activated.

By integrating information across blocks, MoB effectively leverages the representational capacity of all Transformer blocks, thereby accelerating the video generation process without compromising output quality. Furthermore, MoB incorporates a knowledge distillation mechanism to reduce the reliance on large-scale training data, enabling practical adaptation through fine-tuning on smaller datasets.

In terms of the training objective, MoB introduces load balancing strategies into its loss formulation to ensure stable and efficient optimization. These collective design choices empower MoB to produce high-quality outputs with reduced computational overhead. The detailed architecture of **MoB** is shown in Figure 3. During training, the Ada-Top-$k$ mechanism is used to approximate the skipping of block computations. The output of each block is formulated as the sum of its input and the weighted contribution of the block transformation. Since no block is skipped during training, the Block Cache module (Section 2.2) records only the output differences rather than performing full computations, thereby ensuring sufficient training of the routing network.

### 2.1 ROUTING NETWORK

In MoB, the dynamic routing network is placed before the denoising network and takes as input the text embeddings $z_{\text{text}} \in \mathbb{R}^{B \times T \times C}$ from the text encoder, where $B$ is the batch size, $T$ is the token sequence length, and $C$ is the embedding dimension. The routing network first applies average pooling along the $T$-dimension to compress token-level representations into a global representation $z_{\text{cond}} \in \mathbb{R}^{B \times C}$.

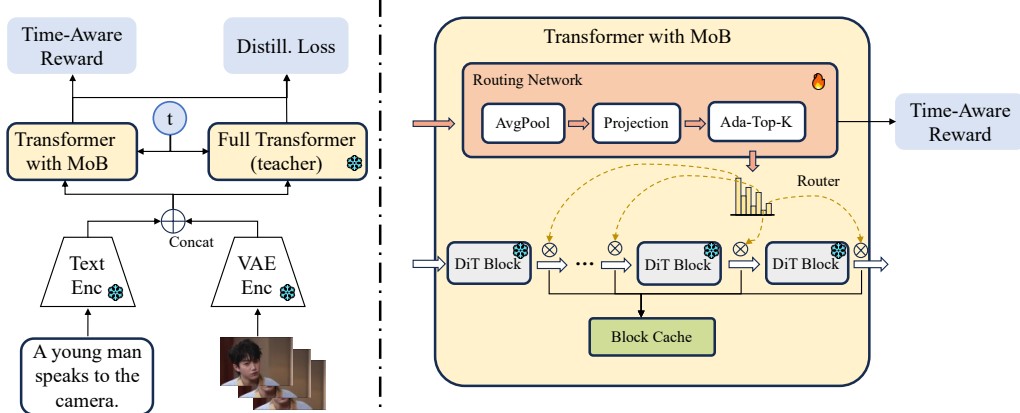

Figure 3: Training Pipeline of MoB. We train only the dynamic routing network while keeping all Transformer block parameters frozen. The routing network employs Ada-Top-$k$ to generate soft selection weights for each block during training, enabling gradient propagation through all blocks. The training objective combines knowledge distillation loss from the teacher model with a time-aware reward term that encourages sparse block activation.

$$z_{\text{cond}} = \frac{1}{T} \sum_{t=1}^{T} z_{\text{text}}[:, t, :] \tag{1}$$

Subsequently, a fully connected layer projects $z_{\text{cond}}$ into a space of dimension equal to the number of Transformer blocks $N$, yielding block-wise importance scores $s \in \mathbb{R}^{B \times N}$, as shown in Equation 2, where $W$ and $b$ are learnable parameters.

$$s = W z_{\text{cond}} + b \tag{2}$$

**Ada-Top-$k$.** In MoB, dynamic routing is performed at the Transformer block level, where each block is executed sequentially. Consequently, the conventional MoE approach—normalizing expert scores $s$ into weights via a softmax operation—is not directly applicable. Such normalization would assign extremely small weights to individual blocks, thereby impeding convergence during training. Conversely, directly applying Top-$k$ or Top-$p$ strategies is problematic, as these operations are non-differentiable and lead to unstable gradient flow during optimization.

The standard Soft-Top-$k$ approximates Top-$k$ selection via softmax with temperature scaling $T_{\text{temp}}$. However, lowering the temperature sharpens the Soft-Top-$k$ toward a one-hot distribution, conflicting with the intended $k$-hot outcome

---

**Algorithm 1** Dynamic Block Routing

**Require:** Text embeddings $z_{text}$; Transformer with $N$ blocks $\{f_1, \ldots, f_N\}$; Router $g$; budget $k$; init $h_0$
**Ensure:** Output $h'$
1: $r \leftarrow g(z_{text})$
2: $h \leftarrow h_0$
3: **for** $n = 1$ to $N$ **do**
4:      **if** $r_n \approx 1$ **then**
5:          $h \leftarrow f_n(h)$
6:      **else**
7:          **continue**
8:      **end if**
9: **end for**
10: $h' \leftarrow h$
11: **return** $h'$

---

To address this, we propose the **Ada-Top-$k$** (Adaptive Top-$k$), a modified soft top-k mechanism. Ada-Top-$k$ computes the k-th largest block importance score $s_{(k)}$, re-centers all block scores relative to this adaptive threshold, and applies a sigmoid activation function:

$$r_i = \frac{1}{1 + \exp(-(s_i - s_{(k)})/T)}, \quad r_i \in [0, 1] \tag{3}$$

where $T$ is the temperature parameter controlling the sharpness of selection. Unlike softmax-based soft top-k which suffers from competitive normalization, Ada-Top-$k$ employs a rank-based formulation that enables independent block activation—blocks with scores above $s_{(k)}$ are activated while

those below are suppressed, naturally selecting approximately k blocks without winner-take-all collapse.

In summary, the dynamic routing network in MoB generates block importance scores conditioned on input prompts, then employs Ada-Top-$k$ to determine block execution. The parameter $k$ controls the target number of active blocks, enabling flexible trade-offs between computational efficiency and generation quality. Algorithm 1 outlines the integration of this routing mechanism into the DiT framework, where $f_n$ denotes the n-th Transformer block, $g$ represents the routing network, and $h_0 \in \mathbb{R}^{B \times T \times C}$ is the initial noise input.

## 2.2 BLOCK CACHE

During inference with the routing network, some blocks are inevitably skipped to accelerate computation, which results in the omission of certain block-level information. To address this and preserve inference quality, MoB introduces a **block cache module**.

As illustrated in Figure 1(b), although different timesteps influence the overall trend of block-level output similarity, the outputs of certain adjacent blocks remain highly consistent. Exploiting this property, we cache the output difference $\delta$ of each block at every timestep. When a block at position $i$ is skipped in the next timestep, the cached $\delta$ from block $i - 1$ at the previous timestep is used to correct its output, as shown in Figure 4.

Formally, this procedure is defined in Equations 4, where $h_i^{(t)}$ denotes the output of the $i$-th block at timestep $t$. Through this caching mechanism, MoB reduces computational cost while maintaining generation performance.

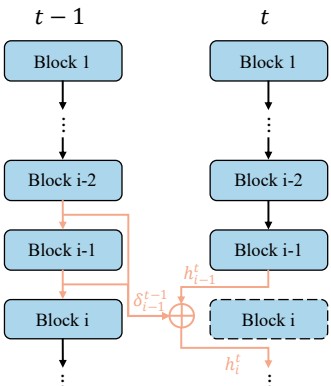

Figure 4: Workflow of the Block Cache Module.

$$h_i^{(t)} \approx h_{i-1}^{(t)} + \left( h_{i-1}^{(t-1)} - h_{i-2}^{(t-1)} \right) \quad (4)$$

## 2.3 OTHER OPTIMIZATIONS

**Timestep Skipping.** In our experiments, we observe that Transformer computations at certain mid-denoising timesteps exhibit substantial redundancy. Removing all computations at these timesteps has little effect on the final generation quality, a finding consistent with prior work that accelerates inference by skipping timesteps (Zhu et al., 2025).

Further analysis shows that, during the mid-denoising stage, the model can maintain stable generation quality provided that multiple skips do not occur consecutively across several timesteps, as illustrated in Figure 5. In the figure, the first row shows the generation results of the original CogVideoX, while the second row corresponds to continuous timestep skipping. The third row applies skipping during the early denoising stages, and the fourth row applies intermittent skipping during the mid-denoising

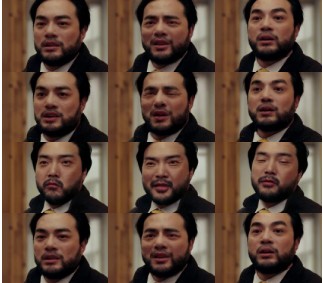

Figure 5: Generation Results of the Original CogVideoX and Three Timestep Skipping Strategies.

stages. Across all settings, the number of skipped timesteps is kept constant. The results show that intermittent skipping in the mid-denoising stages yields the best generation quality.

Motivated by this observation, we introduce an additional **timestep skipping strategy** to further reduce computational cost. Specifically, we define a fixed interval between skipped timesteps and constrain skipping to occur only within a predefined minimum and maximum timestep range, thereby restricting it to the mid-denoising stages.

**Distillation.** The objective of MoB is to accelerate video generation by skipping redundant computations while preserving output quality. Accordingly, the training objective is to align the performance of the MoB-augmented model with that of the original model. To reduce training data requirements

and improve training efficiency, MoB incorporates a knowledge distillation strategy based on the teacher–student paradigm.

In this framework, the original text-to-video model serves as the teacher, providing soft supervisory signals to guide the learning of the student model equipped with MoB. Specifically, the student is optimized with a distillation loss that encourages it to mimic either the final outputs or the intermediate feature representations of the teacher.

$$\mathcal{L}_{\text{distill}} = \frac{1}{B} \sum_{b=1}^{B} \|f_{\text{student}}(\boldsymbol{x_b}) - f_{\text{teacher}}(\boldsymbol{x_b})\|_2^2 \tag{5}$$

where $B$ is the batch size, $\boldsymbol{x_b}$ denotes the $b$-th input sample, and $f_{\text{student}}(\cdot)$ and $f_{\text{teacher}}(\cdot)$ represent the output logits from the student and teacher models, respectively.

**Total Loss.** In this study, MoB employs a distillation loss in place of the original task-specific objective. Since the Transformer backbone is kept frozen, the load balancing loss is not required. To further ensure that MoB achieves sufficient acceleration, we introduce a computation-aware reward term, denoted as $\mathcal{L}_{\text{cost}}$.

$$\mathcal{L}_{\text{cost}} = \lambda \cdot \frac{1}{Z} \sum_{i=1}^{N} p_i \cdot c_i \tag{6}$$

where $\lambda$ is a hyperparameter controlling the strength of the reward, $p_i$ denotes the activation probability of the $i$-th block, and $c_i$ represents the computational cost of that block. The term $Z = \sum_{i=1}^{N} c_i$ serves as a normalization factor.

Specifically, the overall loss function is defined as follows:

$$\mathcal{L}_{\text{total}} = \mathcal{L}_{\text{distill}} + \mathcal{L}_{\text{cost}} \tag{7}$$

where $\mathcal{L}_{\text{distill}}$ measures the discrepancy between the outputs of the student and teacher models, and $\mathcal{L}_{\text{cost}}$ represents the computation-aware cost reward.

## 3 EXPERIMENT

### 3.1 SETUP

**Model.** We evaluate the performance of MoB on the text-to-video task using three state-of-the-art models: CogVideoX-5B, HunyuanVideo (Kong et al., 2024), and Wan2.1 (Wang et al., 2025). For comparison, we adopt MagCache (Ma et al., 2025), Sparse VideoGen (Xi et al., 2025), and FasterCache (Lv et al., 2024) as baselines. In addition, we evaluated the performance of MoB on the text-to-image task using Flux.1 (Labs, 2024).

**Metrics.** Following prior work (Wu et al., 2023), we evaluate both generation quality and inference efficiency using the following metrics. For video quality, we adopt two measures: *Frame Consistency*, computed by extracting CLIP embeddings for each frame and reporting the average cosine similarity across all frame pairs, and *Textual Faithfulness*, quantified by the average ImageReward score(Xu et al., 2023) between each generated frame and its corresponding text prompt. In addition, we also evaluate generation quality using PSNR and SSIM. For image quality, we adopt *Textual Faithfulness*, PSNR and SSIM.

For efficiency, we measure inference latency and report the speedup relative to the baseline model.

**Datasets.** In this study, for evaluation, we randomly sample 1,000 prompts from Lin et al. (2014).

**Training Details.** Our MoB model is trained over 20,000 epochs using 8 NVIDIA A800-SXM4-80GB GPUs, with all Transformer modules kept freezed except for the routing network. During training, the weight of the computation-aware reward $\lambda$ is set to 1e-2. The weight decay is set to 1e-4 , gradient clipping is applied with a threshold of 1.0, and the learning rate is initialized to 2e-5, with a cosine learning rate scheduler applied for gradual decay.

Unlike in inference, the routing network does not discretize gating weights during training. Instead, continuous gating weights are applied to all Transformer block outputs, and their weighted aggregation is used as the final output.

Table 1: Comparison of different acceleration methods on CogVideoX, HunyuanVideo and Wan.

| Method | Efficiency | | Visual Quality | | | |
|---|---|---|---|---|---|---|
| | Latency (s)↓ | Speedup↑ | Smooth↑ | Text↑ | PSNR ↑ | SSIM ↑ |
| **CogVideoX-5B (49 frames, 480P)** | | | | | | |
| CogVideoX-5B ($T = 50$) | 227.8 | 1× | 0.9453 | 0.2971 | - | - |
| FasterCache | 144.1 | 1.58× | 0.9430 | 0.2939 | 13.91 | 0.572 |
| SVG | 134.5 | 1.69× | 0.9272 | 0.2992 | 17.73 | 0.778 |
| Ours | 154.9 | 1.47× | 0.9450 | 0.2972 | 16.01 | 0.734 |
| **HunyuanVideo (81 frames, 720P)** | | | | | | |
| HunyuanVideo ($T = 50$) | 1784.2 | 1× | 0.9686 | 0.3020 | - | - |
| MagCache | 706.5 | 2.52× | 0.9681 | 0.3017 | 17.81 | 0.614 |
| SVG | 1055.8 | 1.69× | 0.9655 | 0.2749 | 18.05 | 0.671 |
| Ours | 1427.2 | 1.25× | 0.9672 | 0.2953 | 16.78 | 0.607 |
| **Wan 2.1 1.3B (81 frames, 480P)** | | | | | | |
| Wan 2.1 ($T = 50$) | 103.6 | 1× | 0.9747 | 0.3005 | - | - |
| MagCache | 54.8 | 1.89× | 0.9743 | 0.3002 | 20.21 | 0.742 |
| SVG | 79.2 | 1.31× | 0.9622 | 0.2976 | 18.63 | 0.711 |
| Ours | 71.8 | 1.44× | 0.9702 | 0.2994 | 18.54 | 0.716 |

**Baselines.** As for video generation, we benchmark MoB against representative baselines, including the vanilla model, the cache-based methods MagCache and FasterCache, and the dynamic sparsity-aware model Sparse VideoGen. As for image generation, we benchmark against Flux.1-dev.

**Parameters.** For all baseline methods, we use their publicly released configurations to ensure fair comparison. Since different models contain varying numbers of Transformer blocks, we set 10% of the total blocks as the baseline for MoB's block-skipping strategy and fix the skip interval to 5. These settings balance inference acceleration with generation quality. All inference tasks on the same backbone are performed under a fixed random seed.

## 3.2 EVALUATION

We assess video generation quality by comparing MoB against all baseline methods. The results are presented in Table 1.

MoB demonstrates strong performance in preserving generation quality. On the CogVideoX-5B-T2V model, it achieves a Frame Consistency score of 0.9450 and a Textual Faithfulness score of 0.2969, indicating that generation quality is effectively maintained under accelerated conditions. Representative visualizations are provided in the supplementary figures.

To assess practical applicability, we conduct an end-to-end performance evaluation on an NVIDIA L20X GPU with CUDA 12.4, comparing MoB against all baselines in terms of latency and speedup. Results show that MoB achieves a 1.2×–1.4× acceleration over the baselines with negligible quality degradation.

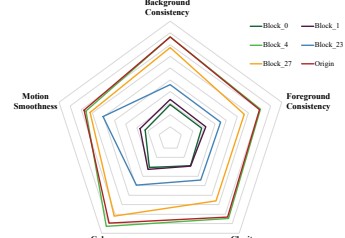

Figure 6: User study on the role of Transformer blocks in CogVideoX-5B.

For image generation quality, we report comparative results in the appendix. Quantitative results are summarized in Table F.1, and representative samples are shown in Figure F.1. MoB achieves superior generation quality — outperforming all baselines with a Text Faithfulness score of 0.3102, a PSNR of 17.64, and an SSIM of 0.7319.

Importantly, the acceleration mechanism of MoB is orthogonal to existing techniques such as attention sparsification. This suggests that MoB can serve as an independent acceleration strategy and can be further combined with methods such as Yuan et al. (2024b) or Jiang et al. (2024) to yield additional efficiency gains in video generation.

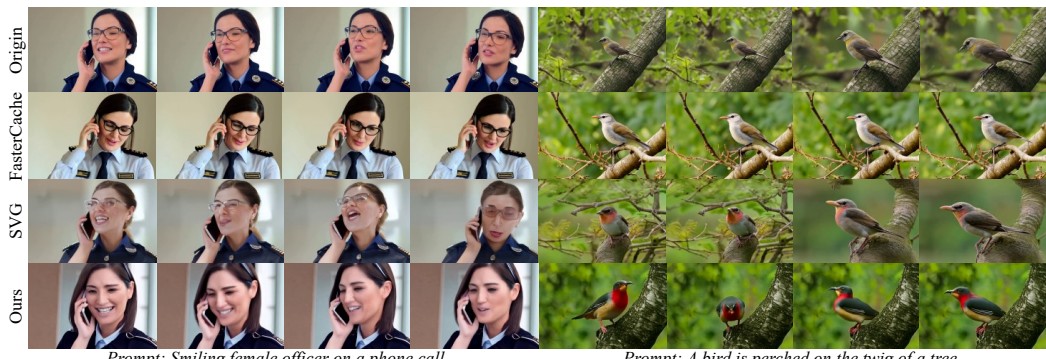

*Prompt: Smiling female officer on a phone call.*     *Prompt: A bird is perched on the twig of a tree.*

Figure 7: Comparison of Generation Results between MoB and Other Baselines on CogVideoX-5B.

## 3.3 USER STUDY

As discussed in previous sections, we observe that different blocks have varying impacts on the output, as measured by MSE. To further qualitatively analyze the role of each block in CogVideoX-5B and to guide the routing network for targeted dynamic routing, we conducted a user study with 100 prompts from Lin et al. (2014).

Participants evaluated videos generated by skipping individual blocks across five aspects: background quality, foreground quality, clarity, color fidelity, and motion smoothness. The results, shown in Figure 6, indicate that different blocks exhibit distinct influences on model performance, while some blocks contribute relatively little to overall generation quality.

## 3.4 ABLATION STUDY

Table 2: Effect of Top-$k$ in routing network

| Top-$k$ | Latency (s)↓ | Speedup↑ | Smooth↑ | Text↑ |
|---|---|---|---|---|
| Origin | 230.1 | 1× | 0.9453 | 0.2958 |
| 40 | 211.3 | 1.08× | 0.9454 | 0.2973 |
| 38 | 194.1 | 1.18× | 0.9431 | 0.2967 |
| 36 | 179.8 | 1.27× | 0.9329 | 0.2817 |

**Effect of Top-$k$ in routing network.** To examine the performance limits of MoB's inference acceleration, we conduct an ablation study, with results summarized in Table 2. The Top-$k$ parameter specifies the number of Transformer blocks selected to participate in the denoising process during inference, as defined in the corresponding equation.

As shown in Table 2, the Top-$k$ setting strongly affects both inference speed and generation quality. While preserving satisfactory visual quality, MoB achieves up to 1.3× speedup. To further illustrate the qualitative impact of varying Top-$k$ values, we present representative visual examples in the accompanying figure, which demonstrate perceptual variations across different configurations.

Table 3: Effect of the Interval Between Two Skips

| Interval | Latency (s)↓ | Speedup↑ | Smooth↑ | Text↑ |
|---|---|---|---|---|
| Origin | 230.1 | 1× | 0.9453 | 0.2958 |
| 2 | 175.1 | 1.34× | 0.9301 | 0.2714 |
| 3 | 195.8 | 1.17× | 0.9316 | 0.2958 |
| 4 | 202.5 | 1.13× | 0.9363 | 0.2957 |
| 5 | 210.3 | 1.09× | 0.9410 | 0.2958 |

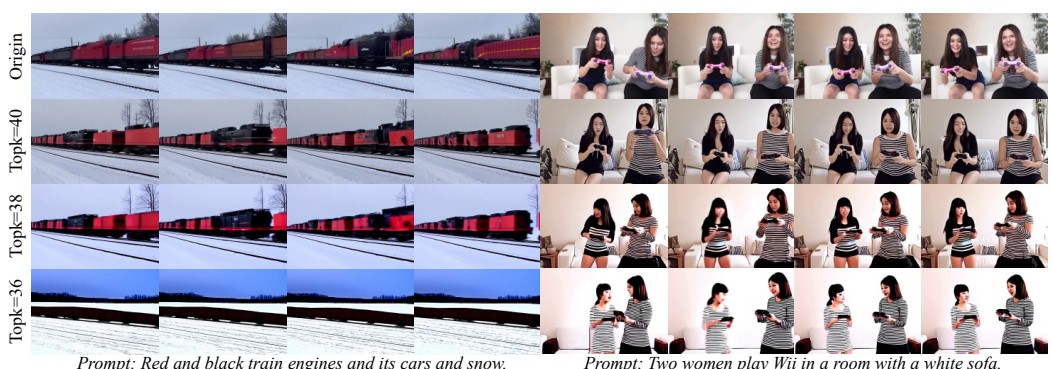

*Prompt: Red and black train engines and its cars and snow.*        *Prompt: Two women play Wii in a room with a white sofa.*

Figure 8: Effect of Top-$k$ in routing network.

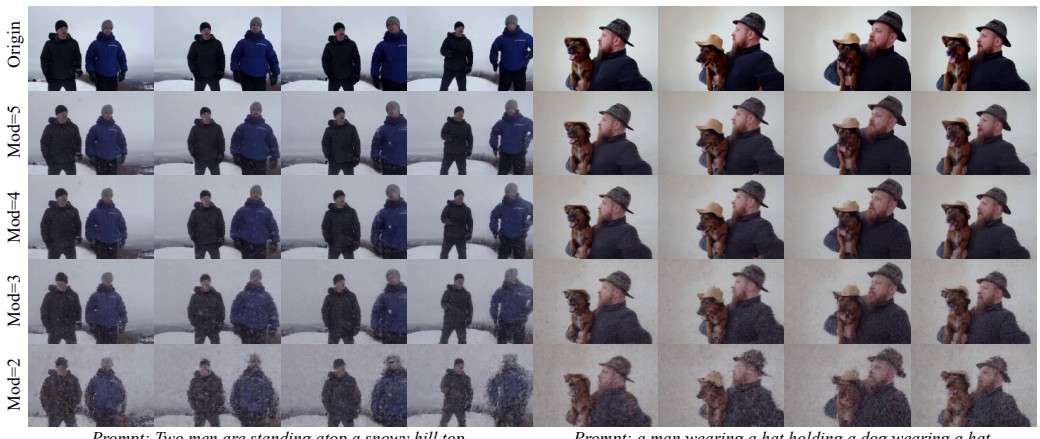

*Prompt: Two men are standing atop a snowy hill top.*        *Prompt: a man wearing a hat holding a dog wearing a hat.*

Figure 9: Effect of the Interval Between Two Skips.

**Effect of the Number of Skipped Timesteps.** To analyze the effect of the timestep-skipping strategy on efficiency and its impact on generation quality, we conduct an ablation study on the number of skipped timesteps. As discussed previously, when skipping occurs in the mid-denoising stage and is applied in a non-consecutive manner, the degradation in video quality is relatively small.

We define the total number of inference steps as $T$. In this experiment, the mid-denoising stage is empirically set to the interval $\left[\frac{3}{10}T, \frac{8}{10}T\right]$. The number of skipped timesteps is controlled by varying the interval between two skips. Experiments are conducted on CogVideoX-5B with $T = 50$, and the results are summarized in Table 3.

**Dynamic Top-$k$ Setting across Timestep.** We evaluate three dynamic Top-$k$ scheduling strategies, including the original scheme, as shown in Figure G.1, to validate the effectiveness of our proposed method.

## 4    CONCLUSION

We propose MoB, a DiT-based acceleration framework for video generation. By exploiting computational redundancy across Transformer blocks, MoB performs block-level dynamic routing to eliminate unnecessary computations. This reduces computational cost while preserving generation capability, and the framework can also be integrated with other acceleration methods.

**Limitations.** MoB trains the routing network based on block-level redundancy. For models with fewer Transformer blocks and stronger computational coupling, training effectiveness may be limited, and block skipping could have a larger negative impact on the final results.

### ACKNOWLEDGMENTS

Use unnumbered third level headings for the acknowledgments. All acknowledgments, including those to funding agencies, go at the end of the paper.

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

# A   ADA-TOP-$k$ ROUTING STRATEGY

As shown in Figure A.1, the weight distributions of three routing strategies are visualized across the 42 Transformer blocks of CogVideoX-5B under the same input. The results show that, while maintaining differentiability, the Ada-Top-k strategy enables better adaptation of the routing network for acceleration. From a mathematical perspective, the Ada-Top-k, Soft-Top-k, and Top-k routing strategies differ primarily in their treatment of normalization and scaling factors. However, Ada-Top-k is the only strategy among the three that is both differentiable and stably yields a k-hot output. During training, given the specific requirements of our block-skipping routing network—particularly its distinction from conventional Mixture-of-Experts (MoE) architectures—we impose two key criteria on the routing mechanism: (1) the routing computation must be differentiable to enable end-to-end training; and (2) the routing output must be k-hot to closely approximate the inference-time scenario in which only a subset of blocks is activated. Given these constraints, we deliberately excluded Top-k (which is non-differentiable) and Soft-Top-k (which yields soft, often near-one-hot distributions rather than exact k-hot selections). Ada-Top-k uniquely satisfies both conditions, making it the most suitable choice for our framework.

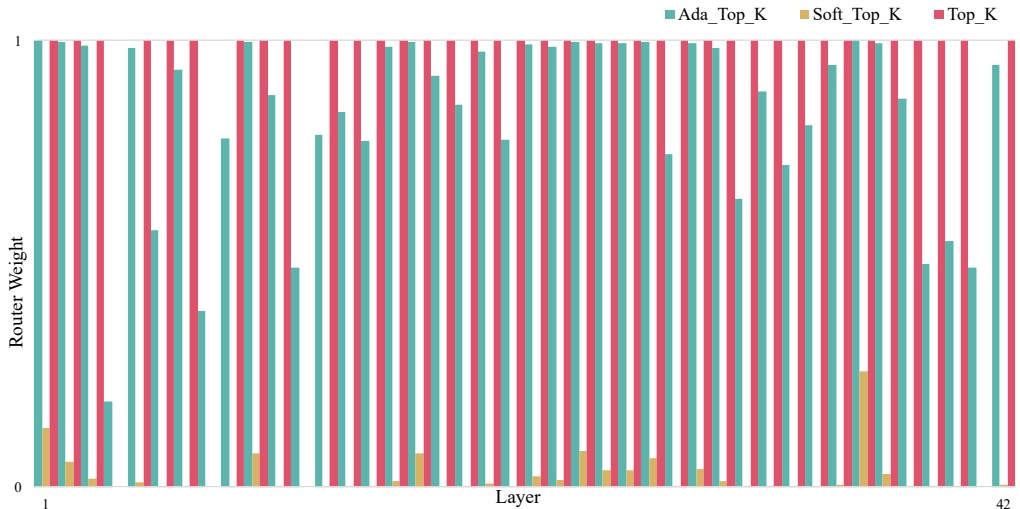

Figure A.1: Output Weight Distributions of Ada-Top-$k$ vs. Other Routing Strategies.

# B   INFERENCE LATENCY AND MEMORY FOOTPRINT OF MOB

In terms of time overhead, using CogVideoX-5B with 42 Transformer blocks as an example, the additional time overhead of the routing network accounts for less than 3% of the original model's inference time. Furthermore, each block skipped by the routing network results in an approximate 2% acceleration.

As for space overhead, the MoB routing network is designed to be lightweight, introducing a negligible number of parameters relative to the original model. During inference, the model caches only the intermediate outputs from the previous timestep, discarding earlier ones to mitigate excessive memory consumption. Consequently, in terms of inference memory overhead, the MoB strategy increases GPU memory consumption by approximately 1%. This efficient resource utilization ensures that the benefits of improved routing are achieved without imposing significant additional memory overhead.

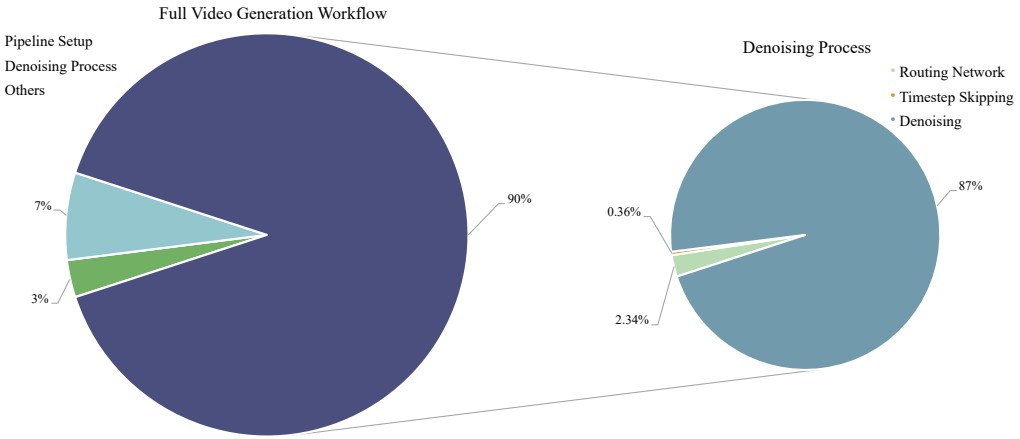

Figure B.1: Time budget allocation across the full inference pipeline with MoB.

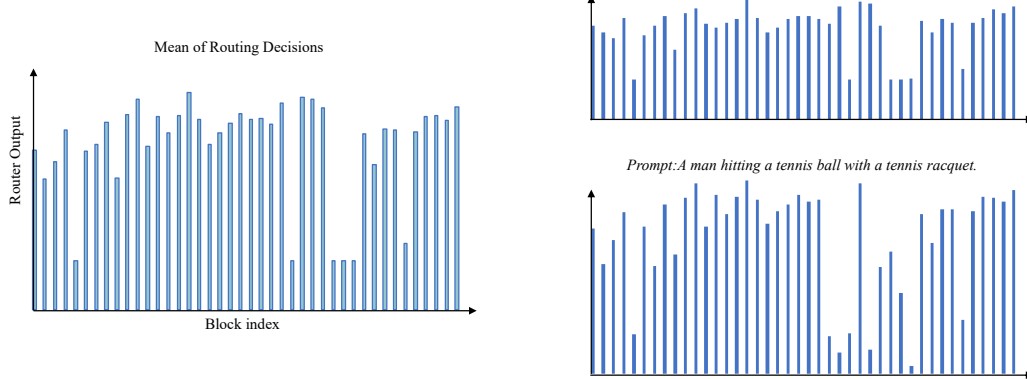

Figure C.1: Visualization of routing decisions.

## C VISUALIZATION OF ROUTING DECISIONS

## D GENERATING LONG-DURATION VIDEOS USING COGVIDEOX-5B W/O MOB

## E STATISTICAL ANALYSIS OF BLOCK-WISE CONTRIBUTIONS IN COGVIDEOX-5B

To analyze the functional contribution of each block, we manually skip individual blocks in CogVideoX when generating videos. Representative cases are shown in Figure E.1.

## F TRANSFER OF MOB TO TEXT-TO-IMAGE GENERATION ON FLUX.1-DEV

We conducted the experiment shown in the Figure F.1 on Flux.1-dev, where the baseline Flux uses 10 denoising steps and contains 19 Transformer blocks. In the MoB-augmented variant, we configure the router to skip 2 blocks, resulting in only 17 active blocks during inference. We further conducted a quantitative evaluation of MoB on Flux.1-dev, comparing its performance with several representative baselines, as shown in Table F.1.

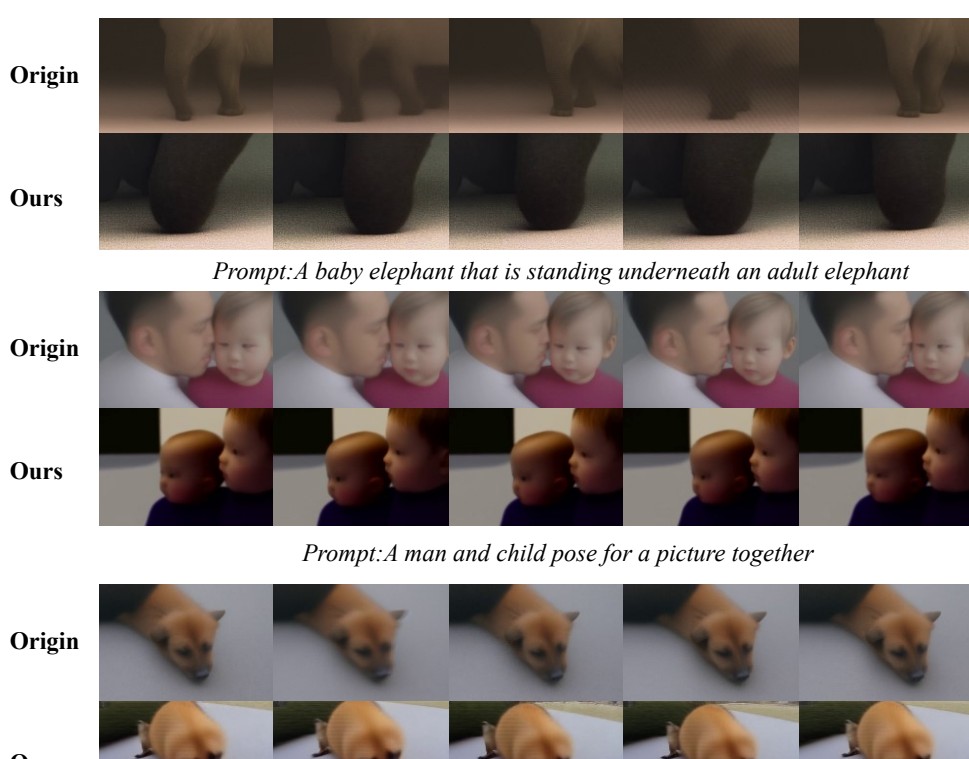

Figure D.1: Generating long-duration videos using CogVideoX-5B w/o MoB.

Table F.1: Evaluation of MoB and Baseline Methods on Flux.1-dev for T2I Acceleration.

| Method | Efficiency | | Visual Quality | | |
|---|---|---|---|---|---|
| | Latency (ms)↓ | Speedup↑ | Text↑ | PSNR↑ | SSIM↑ |
| **Flux.1-dev (1024P)** | | | | | |
| Flux.1-dev ($T = 28$) | 582 | 1× | **0.3108** | - | - |
| MagCache | 277 | **2.10×** | 0.3087 | 17.39 | 0.6952 |
| TaylorSeer(Liu et al., 2025) | 362 | 1.61× | 0.2831 | 13.98 | 0.6179 |
| Ours | 362 | 1.65× | 0.3102 | **17.64** | **0.7319** |

## G  DYNAMIC TOP-K SETTING ACROSS TIMESTEPS

In this section, we investigate whether dynamically varying the Top-$k$ parameter across timesteps can improve generation quality. Figure G.1 compares three routing strategies.

**Random-$k$** refers to performing dynamic routing at each timestep by feeding slightly perturbed inputs into the routing network to induce variation in its outputs. This experiment uses a fixed setting of $k = 40$, as lower Top-$k$ values substantially degrade temporal consistency, making it difficult to assess the effectiveness of the strategy.

**Two-stage-$k$** denotes a strategy in which different Top-$k$ settings are applied across timesteps: during the first half of the denoising process, only 50% of the target number of blocks are skipped, whereas in the second half the full skip ratio is applied. This setup allows us to examine whether delaying aggressive skipping yields better results.

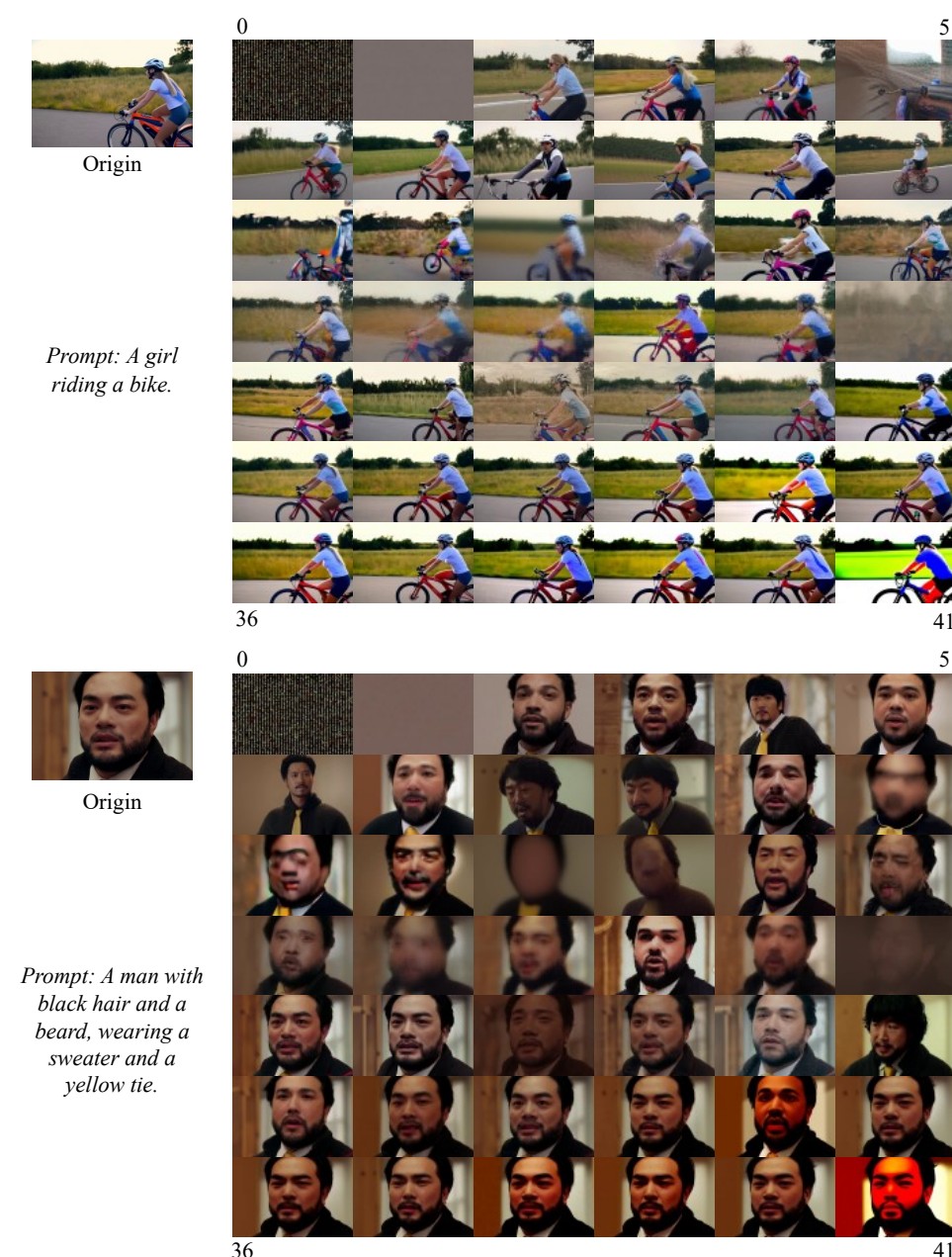

Figure E.1: Visualization of Transformer block outputs across different timesteps.

**Ours** corresponds to the full-pipeline skipping strategy proposed in the main paper, using a fixed $k = 38$ across all timesteps. For a fair comparison, both Two-stage-$k$ and Ours share the same target Top-$k$ configuration.

The results show that skipping different blocks across timesteps leads to noticeable degradation in video consistency. Although the Two-stage-$k$ strategy does not yield significantly worse results, it reduces the total number of block-skipping operations by approximately one quarter, thereby weakening its acceleration effect. As a result, we do not adopt this strategy in our final design.

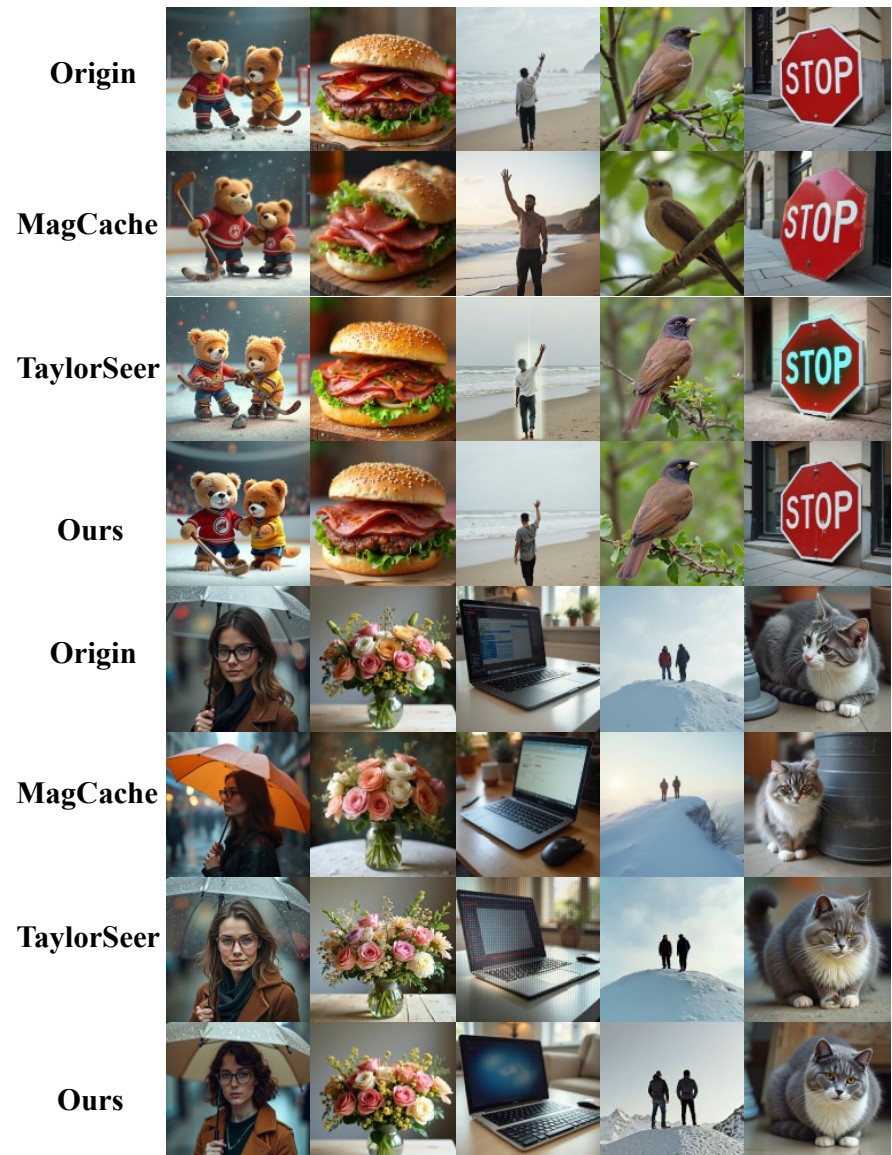

Figure F.1: Comparison of MoB and Baseline Methods on Flux.1-dev.

## H   EVALUATION OF MOB ON HUNYUANVIDEO AND WAN2.1

## I   RELATED WORK

### I.1   DIFFUSION MODELS FOR VIDEO GENERATION

Compared to earlier approaches such as Generative Adversarial Networks (GANs), diffusion models have exhibited superior capabilities in high-quality video generation tasks. Early video diffusion models enhanced pre-trained image diffusion models by incorporating temporal convolution layers and temporal attention mechanisms to improve interframe consistency. For instance, Singer et al. (2022) extended image diffusion models by introducing temporal modeling, leveraging learned coherence along the time axis to adapt text-to-image generation into text-to-video synthesis. Chen et al. (2023) further advanced the field by scaling up the dataset used for video diffusion models, enabling video generation conditioned on various inputs, including text, images, and motion-adaptive signals.

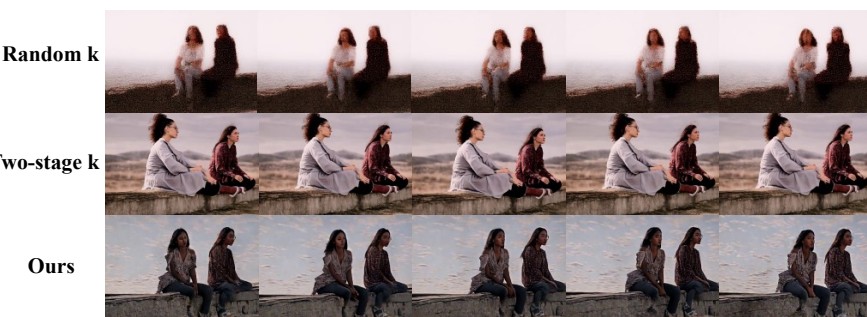

*Prompt: Two women sitting on a ledge deep in thought.*

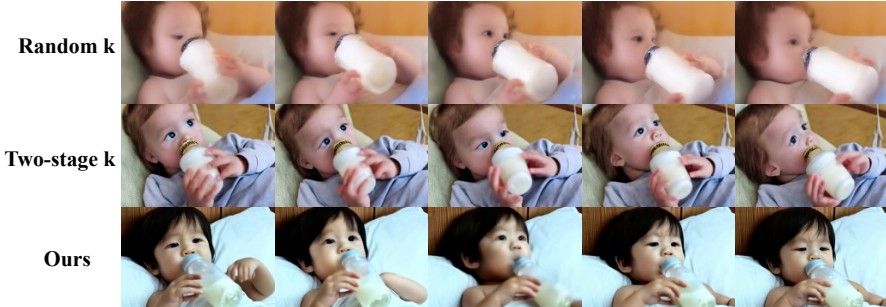

*Prompt: A small child laying in bed drinking a bottle of milk.*

Figure G.1: Comparison of Different Top-k Router Policies.

However, these methods remain constrained by the scalability limitations of the U-Net framework, which makes them less effective in generating longer-duration videos.

In contrast, Diffusion Transformer (DiT)-based models, such as Yu et al. (2023) and Feng et al. (2025), are better suited for capturing complex spatio-temporal dependencies and can generate videos lasting ten seconds or longer. Brooks et al. (2024) further extended DiT for joint spatio-temporal modeling and leveraged large-scale pre-training, achieving the generation of high-quality videos exceeding one minute in length. These advances highlight the advantages of DiT-based models over traditional U-Net-based architectures, particularly in terms of video length and quality.

## I.2 MIXTURE OF EXPERT

In recent years, Transformer-based large language models (LLMs) have exhibited remarkable capabilities, largely driven by their extensive scale, vast training datasets, and substantial computational resources. However, the significant time and financial costs associated with training such models have motivated researchers to explore more efficient strategies to ensure their sustainable development.

The Mixture of Experts (MoE) paradigm was initially introduced in Jacobs et al. (1991) and Jordan & Jacobs (1994). The advent of sparsely gated MoE and its integration into large Transformer-based LLMs have demonstrated its effectiveness in reducing both computational time and memory consumption.

Building upon this idea, Jin et al. (2025b) incorporate this mechanism into the attention module, where each attention head was treated as an expert, and a routing mechanism was employed to dynamically select the Top-$k$ heads for computing weighted attention outputs. Inspired by this, we observe that different DiT blocks contribute unequally to the final generation process. Based on this observation, we propose treating each DiT block as an expert and introducing a routing network that dynamically selects the Top-$k$ blocks for participation in generation, thereby optimizing both training and inference efficiency.

972
973
974
975
976
977
978
979
980
981
982
983
984
985
986

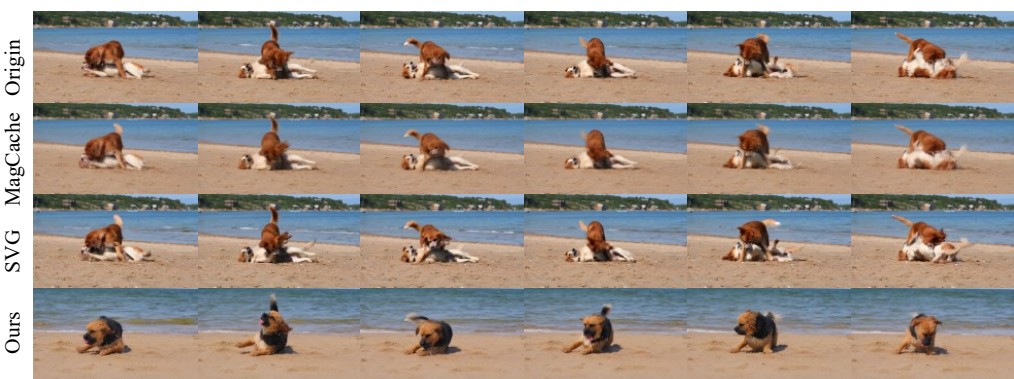

*Prompt: A dog enjoying playing on the beach in the sun.*

987
988
989
990
991
992
993
994
995
996
997
998

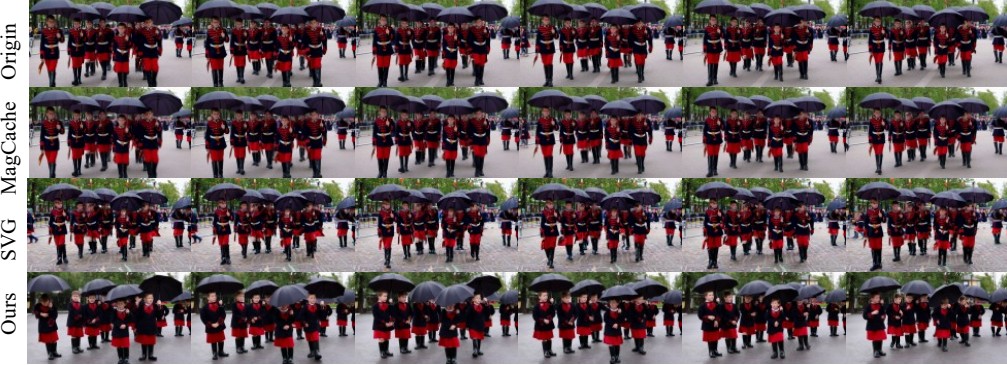

*Prompt: A group of boys wearing black and red uniforms and black boots carries black umbrellas.*

999
1000
1001
1002
1003
1004
1005
1006
1007
1008
1009
1010
1011

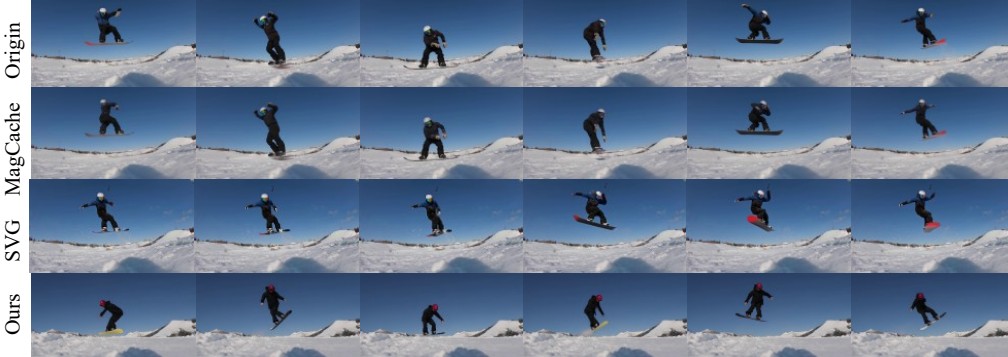

*Prompt: A person in the air on a snowboard.*

1012
1013
1014
1015
1016
1017
1018
1019
1020
1021
1022
1023
1024
1025

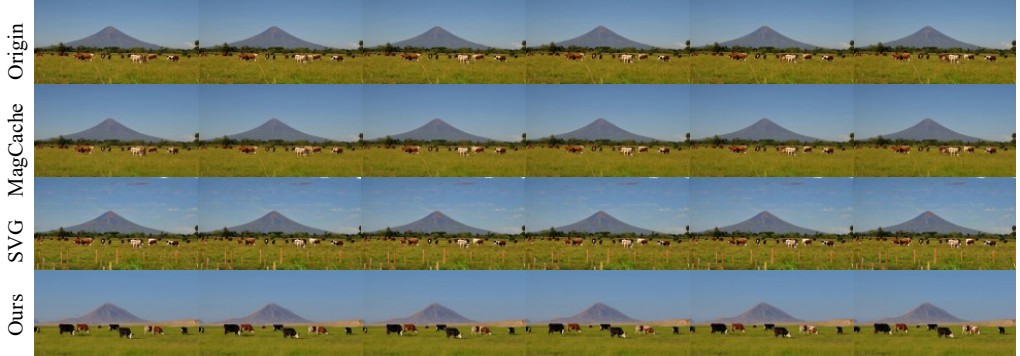

*Prompt: Cows graze peacefully in a field with a dormant volcano in the distance.*

Figure H.1: Results of MoB and Other Acceleration Methods on HunyuanVideo.

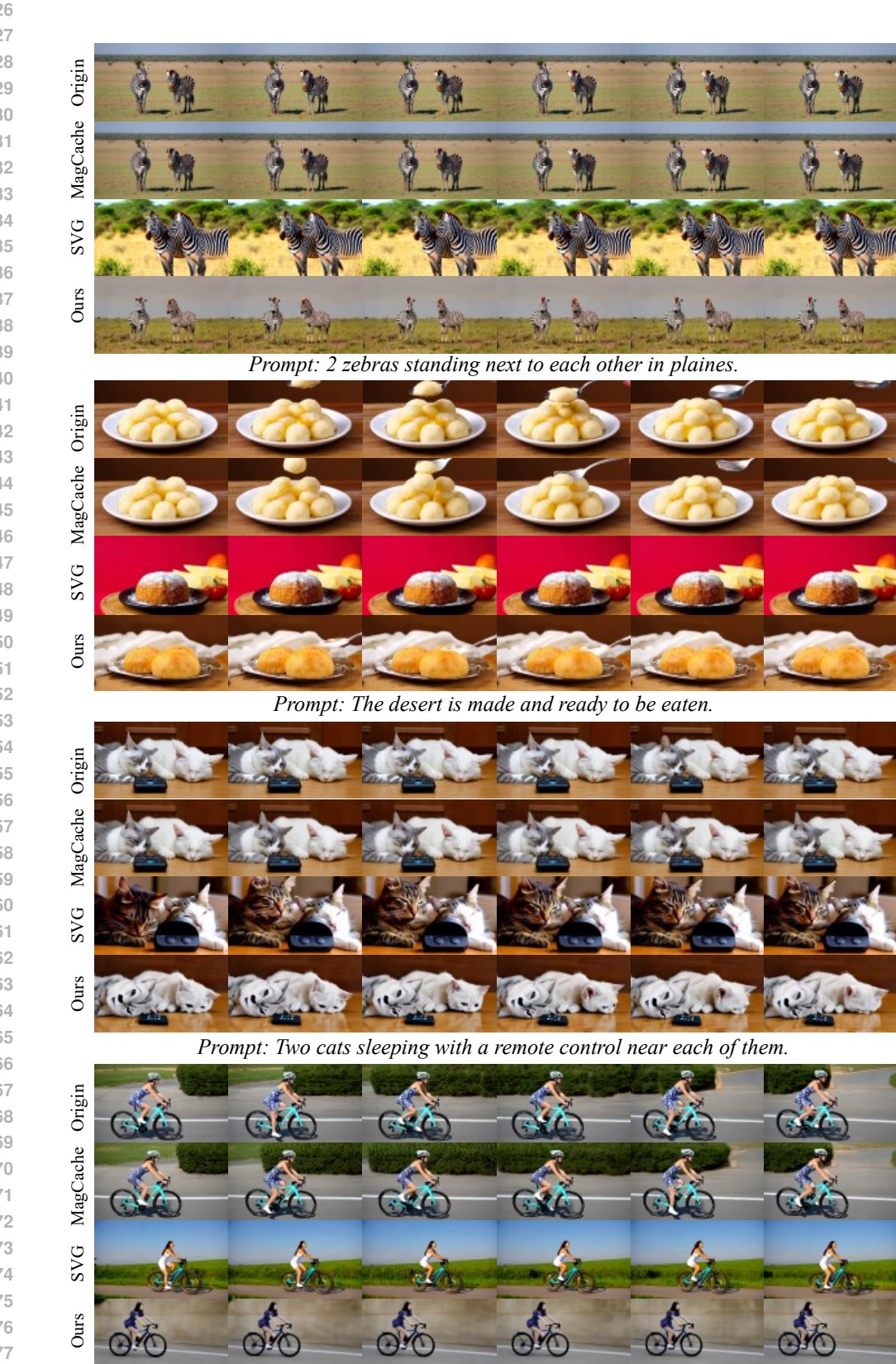

*Prompt: 2 zebras standing next to each other in plaines.*

*Prompt: The desert is made and ready to be eaten.*

*Prompt: Two cats sleeping with a remote control near each of them.*

*Prompt: The woman is riding a bike in a dress.*

Figure H.2: Results of MoB and Other Acceleration Methods on Wan2.1.

