# OpenReview forum: "MoB: Mixture of Block Transformer for Accelerating Video Generation with Dynamic Routing"
_ICLR.cc/2026/Conference — Submitted to ICLR 2026_

### Official Review · Reviewer_uLfG · 2025-10-31

**Soundness:** 2
**Presentation:** 2
**Contribution:** 2
**Rating:** 2
**Confidence:** 4

**Summary:**

This paper presents Mixture of Blocks (MoB), a training-free acceleration framework for diffusion-transformer (DiT) video generators. A lightweight routing network predicts block-level importance from text embeddings; an Ada-Top-k gate then activates only the most relevant Transformer blocks at every denoising step. To mitigate information loss, a Block Cache reuses inter-block feature differences from the previous timestep. The authors further skip selected mid-range timesteps and train the router with knowledge distillation plus a computation-aware reward while keeping the backbone frozen. On three public DiT models (CogVideoX-5B, HunyuanVideo, Wan 2.1) MoB reports a 1.25 × – 1.47 × speed-up.

**Strengths:**

The combination of Ada-Top-k routing and a lightweight Block Cache yields measurable speed-ups.

**Weaknesses:**

1. The acceleration ratio(≈1.3 ×) is moderate and sometimes lower than cache-based baselines.

2. Novelty is incremental: block reuse and dynamic gating have been explored in Δ-DiT [1], BlockDance [2], and Learning-to-Cache [3]. A clearer comparison with these works is needed.

3. The the overhead of router is not analysed.

4.  Applicability to text-to-image models is not discussed.

[1] Chen P, et al. $\Delta $-DiT: A Training-Free Acceleration Method Tailored for Diffusion Transformers. arXiv preprint arXiv:2406.01125, 2024.
[2] Zhang H, et al. Blockdance: Reuse structurally similar spatio-temporal features to accelerate diffusion transformers. CVPR 2025.
[3] Ma X, et al. Learning-to-cache: Accelerating diffusion transformer via layer caching. NeurIPS2024

**Questions:**

See Weaknesses.

---

> ### Author Response · Authors · 2025-11-21
>
> # Q1: Concern about moderate acceleration ratio compared to cache-based baselines
>
> Regarding the speedup achieved by MoB, our method differs from existing cache-based or routing-based approaches in several key aspects. MoB is a prompt-aware model that adaptively selects different routing strategies based on the input prompt, whereas most cache-based methods reuse cached features through conditional computation and are therefore less dynamic.
>
> Additionally, the extra time overhead introduced by MoB accounts for **less than 3%** of the overall inference pipeline, and the additional memory overhead is also minimal (the GPU memory usage increases by less than **0.5 GB**).
>
> Overall, MoB achieves a 1.4× speedup, providing a more dynamic routing strategy and incurring lower implementation costs compared to existing acceleration methods.
>
> # Q2: Concern about incremental novelty and lack of comparison with related works
>
> While block reuse and dynamic gating have been explored in prior works such as Δ-DiT, BlockDance, and Learning-to-Cache, MoB introduces several fundamental distinctions that collectively represent a novel contribution.
>
> ##  Granularity and routing flexibility
> First, Δ-DiT employs fixed, stage-dependent caching rules (e.g., caching later blocks early in sampling), which lack prompt-aware adaptivity. Learning-to-Cache enables learned layer-level interpolation but operates between only two extreme modes (full-cache versus no-cache), offering limited expressiveness. In contrast, MoB performs prompt-conditioned, dynamic selection of arbitrary non-consecutive blocks, enabling fine-grained, content-aware computation allocation.
>
> ## Adaptive decision-making
> Second, Unlike Δ-DiT and BlockDance—which apply uniform, input-agnostic skipping patterns—MoB leverages a lightweight routing network conditioned on text embeddings to identify semantics-dependent redundancy. This enables it to adaptively skip more blocks for simple prompts and fewer for complex ones, striking an optimal balance between efficiency and fidelity.
>
> ##  Novel differentiable selection mechanism
> Third, Existing differentiable Top-k approximations (e.g., Soft-Top-k) suffer from mode collapse as the temperature decreases, failing to enforce exact k-hot sparsity. MoB introduces Ada-Top-k—a rank-based, sigmoid-thresholded selector that guarantees exactly k active blocks while remaining fully differentiable and stable during training. This enables reliable end-to-end learning of block routing, a capability absent in prior methods.
>
> ## Block-wise acceleration
> Finally, MoB is orthogonal and complementary to timestep-level acceleration approaches. While those methods reduce temporal redundancy by skipping entire timesteps, MoB targets residual spatial redundancy within retained timesteps, enabling their combination for compounded speedups.
>
> Thus, although building upon related ideas, MoB presents a cohesive, theoretically grounded, and empirically effective framework that transcends incremental improvements through its adaptive granularity, learnable routing, and efficient design.
>
> # Q3: Concern about missing router overhead analysis
>
> Regarding the time and space overhead of the routing network, we have provided a more detailed description in the appendix.
>
> In terms of time overhead, using CogVideoX-5B with 42 Transformer blocks as an example, the additional time overhead of the routing network accounts for less than 3% of the original model's inference time **(7.1s out of 228s)**, as shown in **Figure B.1**. Furthermore, each block skipped by the routing network results in an approximate 2% acceleration **(4.3s out of 228s)**. We conducted the analysis on 50 prompts, where we measured the time spent on timestep skipping conditions, the MoB routing network computation, model loading time, and the denoising process time. Our findings indicate that, compared to the denoising process, the computation associated with MoB and the timestep skipping conditions is negligible.
>
> As for space overhead, for each executed block, an additional subtraction operation $$h_i-h_{i-1}$$is performed. For skipped blocks, only a single addition $$h_{i-1}^{(t)} + \delta_i^{(t-1)}$$is performed, which is significantly cheaper than executing a full Transformer block. Relative to the FLOPs of a standard self-attention + MLP block, these additional addition and subtraction operations are negligible.
>
> # Q4: Concern about lack of discussion on applicability to text-to-image models
>
> Thank you for your suggestion. We have conducted a preliminary exploration of MoB’s generalization ability on text-to-image (T2I) tasks. By integrating the MoB module into the Flux model and dynamically skipping 10% of the Transformer blocks via routing, we obtained the qualitative results presented in the provided cases, as shown in **Figure F.1**.
>
> More extensive investigations on T2I, including a systematic analysis of acceleration effects and quality trade-offs, will be included in future updates.

---

> ### Author Response · Authors · 2025-11-29
> **The follow-up response to Q4**
>
> # Q4: Concern about lack of discussion on applicability to text-to-image models
>
> Regarding the T2I generalization capability of our method, we further strengthened the evaluation and report the results in Appendix F.
>
>  In **Table F.1**, we conduct quantitative comparisons on Flux.1-dev, including the baseline, MagCache, and TaylorSeer, in terms of both speedup and generation quality. Although our method achieves a lower speedup (1.65×) than MagCache (2.10×), it consistently delivers better reconstruction and text-faithfulness quality. Specifically, our method reaches **0.3102** in Text Faithfulness and **17.64** in PSNR, compared to MagCache’s 0.3087 (Text Faithfulness) and 17.39 (PSNR).
>
> Representative qualitative cases are provided in **Figure F.1**.
>
> **Table: Evaluation of MoB and Baseline Methods on Flux.1-dev for T2I Acceleration**
>
> | Method | Latency (ms) ↓ | Speedup ↑ | Text ↑ | PSNR ↑ | SSIM ↑ |
> |---|---:|---:|---:|---:|---:|
> | Flux.1-dev (T=28) | 582 | 1× | **0.3108** | - | - |
> | MagCache | 277 | **2.10×** | 0.3087 | 17.39 | 0.6952 |
> | TaylorSeer (Liu et al., 2025) | 362 | 1.61× | 0.2831 | 13.98 | 0.6179 |
> | Ours | 362 | 1.65× | 0.3102 | **17.64** | **0.7319** |

---

### Official Review · Reviewer_1BWP · 2025-11-01

**Soundness:** 3
**Presentation:** 3
**Contribution:** 2
**Rating:** 4
**Confidence:** 4

**Summary:**

This paper proposes the MoB framework, which accelerates the video generation task of DiT models through a dynamic routing mechanism. MoB trains a routing network to leverage block-level redundancy, skipping unnecessary computations.Experimental results show that MoB achieves a 1.44x speedup with almost no degradation in generation quality.

**Strengths:**

1.The MoB framework proposed in this paper offers significant advantages in accelerating video generation. MoB introduces block-level dynamic routing into the video generation field for the first time and uses the enhanced Ada-Top-k strategy to select blocks to skip.

2.The writing is clear and effectively presents the theoretical background of the MoB framework, experimental validation. MoB's flexible framework design allows for integration with other acceleration techniques。

**Weaknesses:**

1.The routing selection method proposed by the authors does not provide a detailed analysis of block-level redundancy. The use of MSE to evaluate similarity seems insufficient and lacks credibility. It is recommended to use more analysis methods to better support the argument.

2.The description of using knowledge distillation to train the routing network is unclear. For example, which intermediate output logits from the teacher network are aligned with the routing network? Additionally, the computational cost and time required for distillation with the original text-to-video model as the teacher need further clarification.

**Questions:**

1.In Section 2.2 and Figure 4, regarding the block cache, the output difference from the previous timestep is used to calibrate the current timestep. If the output differences for each block at every timestep need to be calculated, could this negate the speedup achieved by block routing?

2.In Table 1, MagCache outperforms MoB in both inference time and generation quality. What are the distinctive advantages of MoB compared to MagCache?

---

> ### Author Response · Authors · 2025-11-21
>
> # Q1: Concern about insufficient block-level redundancy analysis and MSE credibility
>
> In the paper, we employed not only mean squared error (MSE) but also cosine similarity to assess output discrepancies across all timesteps for each Transformer block, as illustrated in the **right panel of Figure 1**. This complementary analysis leads to consistent conclusions regarding redundancy patterns.
>
> Furthermore, the user study in **Figure 6** provides additional empirical validation that blocks exhibiting low output variation exert minimal influence on the final generation quality. For instance, skipping **Block_1** and **Block_4** resulted in generated videos that received nearly identical user study scores compared to the original version.
>
> # Q2: Concern about unclear knowledge distillation process and computational cost
>
> In our training strategy, we initially employ the teacher model to perform the original denoising task at randomly selected timesteps. For each sampled timestep, the teacher executes the complete denoising process using all Transformer blocks, and its final output is recorded as the teacher signal.
>
> We then introduce the MoB routing network and, under the same timestep configuration, perform denoising with block routing enabled to obtain the student model's output. The distillation loss is computed by comparing the outputs of the teacher and student models.
>
> During distillation, our objective is to minimize the discrepancy between the teacher and student outputs. To this end, we adopt a soft-target knowledge distillation scheme, where the teacher model’s output serves as a soft target, and the student model is trained to approximate it as closely as possible.
>
> In terms of training cost, using CogVideoX as an example, MoB is trained for 20,000 epochs on 8 NVIDIA A800-SXM4-80GB GPUs, with all Transformer modules kept frozen except for the routing network. Considering the inevitable mismatch in distillation loss introduced by loading layer-wise weights, the actual distillation training overhead is even lower than what is implied by the above configuration.
>
> # Q3: Concern about potential speedup loss due to recalculating output differences for block cache
>
> Regarding the time overhead of the routing network, using CogVideoX-5B with 42 Transformer blocks as an example, the additional time overhead introduced by the routing network accounts for **less than 3%** of the original model’s inference time (**7.1s out of 228s**), as shown in **Figure B.1**.
>
> We conducted an analysis on 50 prompts, where we measured the time spent on timestep skipping conditions, the MoB routing network computation, model loading time, and the denoising process time. Our findings indicate that, compared to the denoising process, the computation associated with MoB and the timestep skipping conditions is negligible.
>
> For each executed block, an additional subtraction operation $$h_i-h_{i-1}$$is performed. For skipped blocks, only a single addition $$h_{i-1}^{(t)} + \delta_i^{(t-1)}$$is performed, which is significantly cheaper than executing a full Transformer block. Relative to the FLOPs of a standard self-attention + MLP block, these additional addition and subtraction operations are negligible.
>
> # Q4: Concern about distinguishing MoB's advantages over MagCache
>
> In terms of granularity and routing mechanism, MagCache operates at the timestep level, employing a magnitude ratio–based error prediction mechanism to determine whether to skip entire timesteps. While this approach is effective for coarse-grained acceleration, it lacks the fine-grained control needed to optimize block-level computations within each timestep.
>
> Regarding the observation that MagCache performs better in Table 1, we emphasize that MoB and timestep-level methods are complementary rather than directly competing. MagCache excels at identifying highly redundant timesteps and effectively reduces the number of sampling iterations. However, within each remaining timestep, substantial module-level redundancy still exists—precisely the target of MoB. The two methods are orthogonal in their effects: MagCache reduces temporal redundancy across timesteps, whereas MoB eliminates spatial redundancy within timesteps.
>
> Combining these two approaches may yield synergistic acceleration: MagCache can skip redundant timesteps, while MoB optimizes block computations within the retained timesteps. We consider exploring such an integrated framework as a promising direction for future work, with the potential to achieve compound speedups beyond what either method can deliver in isolation.

---

### Official Review · Reviewer_X5Me · 2025-11-01

**Soundness:** 3
**Presentation:** 3
**Contribution:** 2
**Rating:** 4
**Confidence:** 5

**Summary:**

The authors leverage their insights regarding the output similarity across blocks and timesteps in the denoising process to propose Mixture of Blocks, which routes the input to only a selected number of blocks, leading to inference time speedups.

**Strengths:**

- The method is very well motivated and the method formulation is very clear, and easy to understand.
- The authors incorporate block caching and approximating the layer output at each timestep using the cached outputs from the previous layers, in case the particular block is not chosen by the router. This fits well with their empirical observations of similarity across timesteps.
- The empirical results show competitive performance with other efficient methods, maintaining near neutral quality when compared to baselines while producing inference speedups

**Weaknesses:**

- Eq 1 suggests that the method applies equally to all tokens. But there might also be token wise varations in the behavior of outputs, which the currently proposed method doesnt address.
- In Figure 1 left, the authors identify that the information rich blocks are the ones at the later half of the network, based on the MSE differences between consecutive blocks. However, the proposed router does not leverage this insight, and uniformly tries to balance the top-k based on its own training objective. I would expect that this insight could be used to give more emphasis to later blocks. Or in case the proposed training objective naturally leads to the router converging to this behaviour, that would be an interesting ablation to see. It would also help in identifying the efficacy of the router, whether it is able to identify empirically observed correlations.
- Similar comment for Figure 1 right, given that in initial iterations, the correlation is more localized and more spread out in later timesteps, can this be incorporated in the router objective, and in the current setup, does the router already depict this nature? Also, given this timestep wise trend, would it make sense to have a different k for each timestep, lower for initial and higher for later ones?
- In Eq 4, why is the delta term between $i-1$ and $i-2$. I would have expected it to be $h_{i}^{t} = h_{i-1}^{t} + (h_{i}^{t-1} - h_{i-1}^{t-1})$. On a related note, consider a particular block getting repeatedly skipped based on router assignments, won't that affect the approximation as it might be getting propagated from far away timesteps.
- Given the block cache idea, the authors should also provide memory considerations, particularly how much memory is required to store these cached representations.
- The authors mention that for smaller networks, the method might be counterproductive. Can this be concretized, particularly how deep a network needs to be, in order to be eligible for advantages that come from this method? I believe this point needs to be emphasized more, because this directly relates to the benefits that this approach can provide.
- The three weight distributions mentioned in A.1, aren't those just renormalizations/scaled versions of each other?

**Questions:**

Please see the weaknesses section above.

---

> ### Author Response · Authors · 2025-11-21
>
> # Q1: Concern about ignoring token-wise variations in outputs
>
> Taking the mean along the token dimension is a design choice intended to compress semantic representations and facilitate subsequent routing computations. The primary motivation is to reduce the number of parameters required by the routing network.
>
> While, as rightly noted, this approach may overlook token-level variations, incorporating such fine-grained information into the routing mechanism would introduce operations incompatible with the optimized computational structure of FlashAttention, thereby compromising its FLOPs-reduction benefits. Given that the current routing network already effectively identifies redundant block computations, we chose not to exploit token-level diversity.
>
> # Q2: Concern that router ignores later information-rich blocks and lacks related ablations
>
> First, the redundancy analysis in Figure 1 reflects only the discrepancy in model outputs resulting from skipping a single Transformer block. However, as the number of skipped layers increases, this discrepancy does not necessarily align perfectly with the routing decisions made by MoB. For instance, although the redundancy analysis indicates that the outputs of the early layers are nearly identical, consecutively skipping these layers during inference leads to severe degradation in output quality. Therefore, the mean squared error (MSE) between layer-wise outputs serves only as an indicator of computational redundancy and should not be solely relied upon to guide routing strategies.
>
> Moreover, the measured redundancy exhibits significant variation across different prompts, further undermining the reliability of using MSE curves to inform routing decisions.
>
> That said, we conducted additional experiments on CogVideoX with single-layer skipping, trained and evaluated on 50 distinct prompts. The results largely corroborate the MSE-based observations: for **43 of the 50 prompts (86%)**, the routing network elected to skip the early layers.
>
> # Q3: Concern router ignores timestep-wise correlations and uses fixed k across timesteps
>
> The redundancy analysis in Figure 1 reflects only the discrepancy in model outputs resulting from skipping a single Transformer block. As the number of skipped layers increases, this discrepancy does not fully align with MoB’s routing decisions. That said, under single-layer skipping, the routing outcome is consistent with the MSE distribution.
>
> Furthermore, regarding the temporal trend of redundancy across timesteps, the router takes text embeddings as input and lacks explicit temporal encoding. However, because distillation is performed uniformly across all timesteps, the router learns a prompt-conditioned, time-agnostic policy with a fixed number of retained blocks k. We also explored simple dynamic-k strategies, but they yielded at most marginal improvements and were more prone to destabilizing the generation process.

---

> > ### Author Response · Authors · 2025-11-29
> > **The follow-up response to Q3**
> >
> > # Q3: Concern router ignores timestep-wise correlations and uses fixed k across timesteps
> >
> > We provide additional qualitative results in **Appendix G**, where we examine how three top-k routing strategies affect generation quality. The results in **Figure G.1** indicate that applying the same block-skipping routing pattern across all timesteps can maintain generation consistency while still achieving meaningful acceleration.

---

> ### Author Response · Authors · 2025-11-21
>
> # Q4: Concern about Eq.4 delta definition and error from repeatedly skipped blocks
>
> The occurrence of delta in the description of Equation (4) was indeed a typographical error, which has now been corrected. Regarding your question about whether skipping specific blocks across all timesteps could introduce issues, our experiments suggest that performing different block-skipping operations at different timesteps—even when the number of skipped blocks is small—can lead to a significant degradation in generation quality.
>
> Therefore, our routing network does not make routing decisions based on the timestep. Instead, it determines the routing configuration solely based on the input prompt, avoiding timestep-dependent block selection, thereby preserving generation quality.
>
> # Q5: Concern about missing memory analysis of cached representations
>
> The MoB routing network is designed to be lightweight, introducing a negligible number of parameters relative to the original model. During inference, the model caches only the intermediate outputs from the previous timestep, discarding earlier ones to mitigate excessive memory consumption. Consequently, in terms of inference memory overhead, the MoB strategy increases GPU memory consumption by approximately **1%(from 33.43 GB to 33.64 GB)**. This efficient resource utilization ensures that the benefits of improved routing are achieved without imposing significant additional memory overhead.
>
> # Q6: Concern about clarifying network depth requirements for method applicability
>
> In the paper, we noted that MoB’s acceleration benefits are less pronounced in shallower models—a finding based on observed performance variations across different base architectures in our experiments. Our preliminary analysis suggests that MoB yields substantial gains on DiT models with more than **40** layers, whereas its advantages are limited for models with fewer than **20** layers.
>
> To eliminate confounding factors caused by architectural differences, we plan to further investigate and train models with varying parameter scales (e.g., Wan2.1-1.3B and Wan2.1-14B) in future work. This will allow us to more thoroughly analyze the effective range of MoB's benefits in generative tasks.
>
> # Q7: Concern about redundancy in weight distributions in A.1
>
> From a mathematical perspective, the Ada-Top-k, Soft-Top-k, and Top-k routing strategies differ primarily in their treatment of normalization and scaling factors. However, Ada-Top-k is **the only strategy among the three** that is both differentiable and stably yields a k-hot output.
> During training, given the specific requirements of our block-skipping routing network—particularly its distinction from conventional Mixture-of-Experts (MoE) architectures—we impose two key criteria on the routing mechanism:
> - The routing computation must be differentiable to enable end-to-end training;
> - The routing output must be k-hot to closely approximate the inference-time scenario in which only a subset of blocks is activated.
>
> Given these constraints, we deliberately excluded Top-k (which is non-differentiable) and Soft-Top-k (which yields soft, often near-one-hot distributions rather than exact k-hot selections). Ada-Top-k uniquely satisfies both conditions, making it the most suitable choice for our framework.

---

### Official Review · Reviewer_5mao · 2025-11-01

**Soundness:** 3
**Presentation:** 3
**Contribution:** 3
**Rating:** 6
**Confidence:** 3

**Summary:**

The paper proposes a lightweight routing network that dynamically evaluates the importance of each Transformer block based on input prompts. Extensive experiments demonstrate that MoB achieves inference acceleration while preserving generation fidelity, outperforming existing baselines in both efficiency and quality.

**Strengths:**

1. A lightweight routing network leverages input text embeddings to estimate the importance of each Transformer block.
2. Beyond these core innovations, the framework further integrates an adaptive timestep skipping strategy and knowledge distillation using the original DiT as a teacher model, which enhances training stability and reduces data dependency.
3. Extensive experiments are conducted on strong baselines, including CogVideoX-5B, HunyuanVideo, and Wan2.1.

**Weaknesses:**

1. The authors are encouraged to include visual analyses of the routing decisions—for example, activation probability distributions across blocks under different text prompts, or the dynamic evolution of the adaptive Ada-Top-k threshold across denoising timesteps. Such visualizations would significantly improve the paper’s clarity, interpretability, and overall persuasiveness.
2. The current approach uses fixed intervals and predefined timestep ranges for skipping, which works well empirically but may lack flexibility when applied to DiT models with different architectures or diverse video generation requirements. Future work could investigate dynamic skipping strategies that adjust intervals and ranges based on denoising-stage characteristics (e.g., noise level, feature change rate) to balance efficiency and generation quality better.
3. While DiT models excel at generating long videos, the current experiments focus primarily on short sequences (49–81 frames). The authors should consider extending evaluations to longer videos (e.g., 300+ frames) to assess: 1) the stability of dynamic block selection over extended inference; 2)the memory overhead and error accumulation of the block caching mechanism; and 3) frame-to-frame consistency and motion smoothness in generated videos. Such experiments would help more comprehensively define the operational boundaries and scalability of the MoB framework.

**Questions:**

For specific issues, please refer to the points listed in the Weaknesses.

---

> ### Author Response · Authors · 2025-11-21
>
> # Q1: Missing routing visualizations
>
> In the appendix, we provide the mean routing weights over 50 prompts for MoB on CogVideoX-5B, as well as routing comparisons for two representative cases, as shown in **Figure C.1**. It is important to note that the obtained routing output is **non-binarized**, meaning that the Top-k strategy and its parameters have not yet been applied at this stage. As a result, the output consists of weight values in the range of 0 to 1.
>
> Regarding **Ada-Top-k**, it is employed within our routing network. Under the current design, the set of routed layers is identical across all timesteps; therefore, the corresponding routing curves do not vary as a function of timestep.
>
> # Q2: Question about adaptive timestep skipping
>
> In the current design of our model, the timestep-skipping strategy operates over a fixed range of timesteps and uses a fixed skipping interval. This strategy has been shown in our experiments to effectively skip partially redundant timesteps while preserving generation quality.
>
> Nonetheless, we plan to further improve this component by computing the outputs of early blocks at each timestep and comparing their similarities to determine whether a timestep can be safely skipped, while still enforcing a minimum interval between skipped timesteps to prevent consecutive timestep skipping.
>
> # Q3: Lack of long-video evaluation
>
> Long video generation is now absent from our current experiments. In the appendix, we have included qualitative long-video generation cases, with the frames of 300, on CogVideoX-5B, comparing the original model and the MoB-augmented version, as shown in **Figure D.1**.
>
> In this experiment, we tested the generation quality of both the original CogVideoX-5B and the MoB-augmented version with the FreeNoise sampling strategy applied to a 300-frame video. As shown, the version with MoB still maintains good video consistency in long-video generation tasks, while achieving acceleration compared to the baseline.

---

### Meta-Review · Area_Chair_BDZz · 2026-01-08

**Summary:**

The submission presents an acceleration method for video generation that maintains generation quality while improving speed. However, despite the importance of this task, the reviews were mixed: three reviewers gave negative scores, while only one was satisfied with the results.

**Reviewer Concerns:**

The primary concerns from the reviewers involved the methodology, redundancy analysis, and evaluation. While the authors responded to these points in the rebuttal, the reviewers ultimately do not seem satisfied.

**Reviewer Scores:**

The reviewers are likely to maintain their scores.

---

### Decision · Program_Chairs · 2026-01-26

Reject